# Lutein Decreases Inflammation and Oxidative Stress and Prevents Iron Accumulation and Lipid Peroxidation at Glutamate-Induced Neurotoxicity

**DOI:** 10.3390/antiox11112269

**Published:** 2022-11-17

**Authors:** Ramóna Pap, Edina Pandur, Gergely Jánosa, Katalin Sipos, Tamás Nagy, Attila Agócs, József Deli

**Affiliations:** 1Department of Pharmaceutical Biology, Faculty of Pharmacy, University of Pécs, Rókus u. 2, H-7624 Pécs, Hungary; 2Department of Laboratory Medicine, Faculty of Medical Sciences, University of Pécs, Ifjúság út 13, H-7624 Pécs, Hungary; 3Department of Biochemistry and Medical Chemistry, Medical School, University of Pécs, Szigeti út 12, H-7624 Pécs, Hungary; 4Department of Pharmacognosy, Faculty of Pharmacy, University of Pécs, Rókus u. 2, H-7624 Pécs, Hungary

**Keywords:** neuron, lutein, glutamate, iron, oxidative stress, cytokines

## Abstract

The xanthophyll carotenoid lutein has been widely used as supplementation due to its protective effects in light-induced oxidative stress. Its antioxidant and anti-inflammatory features suggest that it has a neuroprotective role as well. Glutamate is a major excitatory neurotransmitter in the central nervous system (CNS), which plays a key role in regulating brain function. Excess accumulation of intracellular glutamate accelerates an increase in the concentration of reactive oxygen species (ROS) in neurons leading to glutamate neurotoxicity. In this study, we focused on the effects of glutamate on SH-SY5Y neuroblastoma cells to identify the possible alterations in oxidative stress, inflammation, and iron metabolism that affect the neurological function itself and in the presence of antioxidant lutein. First, ROS measurements were performed, and then catalase (CAT) and Superoxide Dismutase (SOD) enzyme activity were determined by enzyme activity assay kits. The ELISA technique was used to detect proinflammatory TNFα, IL-6, and IL-8 cytokine secretions. Alterations in iron uptake, storage, and release were followed by gene expression measurements and Western blotting. Total iron level detections were performed by a ferrozine-based iron detection method, and a heme assay kit was used for heme measurements. The gene expression toward lipid-peroxidation was determined by RT-PCR. Our results show glutamate changes ROS, inflammation, and antioxidant enzyme activity, modulate iron accumulation, and may initiate lipid peroxidation in SH-SY5Y cells. Meanwhile, lutein attenuates the glutamate-induced effects on ROS, inflammation, iron metabolism, and lipid peroxidation. According to our findings, lutein could be a beneficial, supportive treatment in neurodegenerative disorders.

## 1. Introduction

Lutein is a xanthophyll carotenoid, which has been widely used as supplementation because of its antioxidant and anti-inflammatory benefits [1]. Lutein is known to have protective effects against light-induced oxidative damage, used as a prevention against the development of age-related macular degeneration and cataracts [2,3]. Additionally, lutein is suggested to be neuroprotective according to its anti-inflammatory and antioxidant features [4,5,6].

Glutamate (Glu), the crucial excitatory neurotransmitter in the central nervous system (CNS), plays a key role in regulating brain function [7,8]. Glu is incorporated into regulating nerve function as a neurotransmitter [7]. Immoderate accumulation of intracellular Glu accelerates an increase in the concentration of ROS in neurons [9]. Oxidative stress is the leading mechanism of Glu neurotoxicity [10]. Based on the literature, Glu induces neurotoxicity and ROS production in various neurons in a concentration-dependent manner, suggesting that the harmful effect of Glu is associated with oxidative stress [11,12].

The SH-SY5Y human neuroblastoma cell line serves as an in vitro neuronal model for neurodegenerative disorders [13]. Neuroinflammation is an inflammatory response which is induced by disease, stress, infection, or damage and mediated by cytokines such as Tumor Necrosis Factor alpha (TNFα), Interleukin (IL)-6, IL-1β, Interferon (IFN)γ and IL-8, chemokines, ROS, and secondary messengers [14,15,16]. ROS in CNS plays a critical role in regulating inter alia inflammatory responses, iron homeostasis, apoptosis, ferroptosis, and lipid peroxidation [17,18].

Neurodegeneration and oxidative stress among others are characterized by dysfunctional iron homeostasis and lipid homeostasis [19,20,21]. Iron is an essential element that plays a fundamental role in the regulation and development of the CNS, besides participating in several neuronal functions [22]. Iron homeostasis is precisely and tightly regulated in the brain. The consequences of iron dysregulation, including both iron overload and deficiencies, are injurious to CNS [23,24]. Iron-dependent lipoxygenases are suspected to play a pivotal role in inducing lipid peroxidation [25]. Long-chain fatty acid-CoA ligase 4 (ACSL4), Arachidonate 5-Lipoxygenase (ALOX5), and Arachidonate 15-Lipoxygenase (ALOX15) are participated in the fate of cells through the activation of lipid metabolism and promoting ferroptosis [26]. Overexpression of lipid-peroxidizing enzymes is related to several diseases, including diabetes, systemic sclerosis, atherosclerosis, prostate cancer, and colorectal cancer, due to its inflammatory-inducing effect known to be related to neuroinflammation [27,28,29,30].

In our study, we focused on the effects of Glu in vitro on SH-SY5Y neuroblastoma cells and how the possible alterations in oxidative stress, inflammation, and iron metabolism affect the neurological function itself and in the presence of antioxidant lutein.

## 2. Materials and Methods

### 2.1. Isolation of Lutein

Lutein (Figure 1) was isolated in the same way as in our previous article [5]. Diethyl ether was used to dissolve the marigold extract in (INEXA C.A., Ecuador) and it was saponified overnight with 30% potassium hydroxide (KOH) in methanol. Next, the ethereal solution was washed five times with water and thereafter dried and evaporated. The crude saponified extract was crystallized from hexane/toluene to deliver lutein with 98% purity. According to HPLC on a C30 column, lutein contained 6% zeaxanthin. Dimethyl sulfoxide (DMSO, Sigma-Aldrich Kft., Budapest, Hungary) was used as a carrier solvent for lutein; it was diluted at a concentration of 1 mg/mL. The diluted stock was sterile filtered and stored at −20 °C protected from light.

### 2.2. Cell Culture and Treatments

SH-SY5Y human neuroblastoma cells (ATCC, CRL-2266) were maintained and cultured in Dulbecco’s Modified Eagle Medium/Nutrient Mixture F12 (DMEM/F-12; Lonza Ltd., Basel, Switzerland) supplemented with 10% fetal bovine serum (FBS; EuroClone S.p.A, Pero, Italy) and 1% Penicillin/Streptomycin (P/S; Lonza Ltd., Basel, Switzerland) in a humidified atmosphere containing 5% CO_2_ at 37 °C. For the experiments, SH-SY5Y cells were seeded onto 6-well plates, onto 60 × 15 mm dishes, or into 25 cm^2^ flasks in an antibiotic-free complete growth medium and were rested for 24 h before treatments in all experiments. In the preliminary experiments, the cells were treated for 24 h with 2.5 ng/µL, 5 ng/µL, 7.5 ng/µL or 10 ng/µL of lutein; or the cells were treated with 1 mM, 2 mM, 3 mM, 4 mM, 5 mM, 6 mM, 7 mM, 8 mM, 9 mM, 10 mM, 15 mM, or 20 mM of Glu (Merck Life Science Kft., Budapest, Hungary). In additional experiments, SH-SY5Y cells were treated with 10 ng/µL of lutein and/or with 1 mM or 5 mM of Glu. Untreated cells served as absolute controls. DMSO treatments in appropriate concentrations were used as controls of lutein and Glu-treated cells. We used the following abbreviations for the treatments: C signs the proper absolute control at 24 h, 48 h, and 72 h. For further DMSO controls an equal amount of DMSO as in the lutein or in the Glu treatments was added to the cells. SH-SY5Y cells which were treated with Glu and lutein together received the treatments at the same time, as lutein first, which was followed by Glu. Each experiment was repeated in triplicate.

### 2.3. Cell Viability Assay

SH-SY5Y cells were plated onto 96-well plates using 10,000 cells/well. In the viability assays, the cells were treated with lutein or Glu in a time- and concentration-dependent manner. In the case of Glu, we used the concentrations 1 mM, 2 mM, 3 mM, 4 mM, 5 mM, 6 mM, 7 mM, 8 mM, 9 mM, 10 mM, 15 mM, or 20 mM of Glu. In the lutein viability tests 2.5 ng/µL, 5 ng/µL, 7.5 ng/µL, or 10 ng/µL of lutein was used as treatment. The incubation period lasted 6 h, 24 h, 48 h, or 72 h. The viability of SH-SY5Y cells was determined by using a Resazurin based In Vitro Toxicology Assay Kit (Sigma-Aldrich Kft., Budapest, Hungary). An equal amount of DMSO was used for the DMSO controls as in the Glu or lutein treatments. Each treatment was compared to its own appropriate controls and untreated cells were used as the control for the DMSO treatment. The abbreviations of our treatments are represented in Table 1. In each case, 20 µL of Resazurin reagent was added to the cells after the treatments, and the plates were incubated for 2 h at 37 °C, 5% CO_2_. MultiSkan GO microplate spectrophotometer (Thermo Fisher Scientific Inc., Waltham, MA, USA) was used for the absorbance measurements at 600 nm. The viability was expressed as the percentile of the total cell number of the proper control cells.

### 2.4. Detection of the Reactive Oxygen Species

Measurement of oxidative stress was carried out with a Fluorometric Intracellular ROS Kit (Deep red) (Merck Life Science Kft., Budapest, Hungary). To generate ROS, 1 mM, and 5 mM of Glu were used in vitro in the SH-SY5Y cells. In the experiments, 5 × 10^3^ cells/well were plated onto a 96-well plate and rested for 24 h before treatments. Next, 10 ng/µL of lutein was used for the SH-SY5Y cells to examine the effect of lutein on ROS production alone and with the Glu treatments. In the mutual treatment, lutein was added first then immediately followed by the addition of Glu. The kit is able to detect the ROS in living cells. Each experiment has been completed according to the instruction of the manufacturer. Briefly, after the 24 h of resting, the cells were treated with 10 ng/µL of lutein and/or with the different concentrations of Glu to induce ROS production then the plates were incubated for 10, 20, and 30 min at 37 °C with 5% CO_2_. Thereafter the cells were stained with 100 µL ROS deep red dye solution per well and were incubated for 30 min at 37 °C and 5% CO_2_. EnSpire Multimode microplate reader (PerkinElmer, Rodgau, Germany) was used for the absorbance measurements with bottom read mode at 650 nm excitation and 675 nm emission wavelengths. The alteration in ROS was determined as a percentage of control.

### 2.5. Detection of Catalase Activity

The measurement of CAT activity in the SH-SY5Y cell lysates was determined by using a Spectrophotometric Catalase Assay Kit (Merck Life Science Kft., Budapest, Hungary). SH-SY5Y cells were treated similarly as described earlier, in 60 mm cell culture dishes (1 × 10^6^ cells/dish). After the 24 h, 48 h, and 72 h treatments the adherent cells were collected with a scraper and centrifugated at 1000× *g* for 10 min at 4 °C. The cell pellets were homogenised in cold buffer (50 mM potassium phosphate, pH 7.0, containing 1 mM ethylenediaminetetraacetic acid (EDTA)) on ice and centrifugated at 10,000× *g* for 15 min at 4 °C. The collected supernatant was stored on ice for the assay. The measurements were carried out according to the manufacturer’s protocol. Briefly, 12 µL of freshly prepared 1 mM H_2_O_2_ was added into each well (samples, positive control, and sample High Control (HC)) to start the reaction. The flat-bottom 96-well was incubated at 25 °C for 30 min. Then 10 µL of stop solution was added into each sample and positive control well to stop the reaction. Next, 50 µL of Developer Reaction Mix was added to the wells, then mixed and incubated at 25 °C for 10 min. In the assay, catalase first reacts with H_2_O_2_ to produce water and oxygen. The unconverted H_2_O_2_ subsequently reacts with the probe to produce a chromogenic product, which was measured calorimetrically at 540 nm using MultiSkan GO microplate spectrophotometer (Thermo Fisher Scientific Inc., Waltham, MA, USA). The calculation of the catalase enzyme activity of the sample was done by using the catalase activity equation. One unit refers to the amount of enzyme that decomposes 1 µmol of H_2_O_2_ per minute at pH 4.5 at 25 °C (nmol/min/mL = mU/mL). The CAT activity of the SH-SY5Y cells was detected from three independent experiments, each was performed in triplicate.

### 2.6. Detection of Superoxide Dismutase Activity

Superoxide Dismutase Activity Assay Kit (Merck Life Science Kft., Budapest, Hungary) was used for the detection of SOD activity. The kit provides the determination of SOD activity in units of activity (units/mL). SH-SY5Y cells were treated identically as described earlier, in 60 mm Petri cell culture dishes (1 × 10^6^ cells/dish). The cells were collected after treatments by using a cell scraper. Each experiment was performed according to the manufacturer’s instructions. Briefly, SH-SY5Y cells were lysed in ice-cold 0.1 M Trizma-HCl (pH 7.4, containing 0.5% Triton X-100, 5 mM mercaptoethanol, and protease inhibitors). After the lysate was centrifuged at 4 °C (14,000× *g*) for 5 min and the supernatant was transferred into a new tube. The supernatant contains total SOD activity both from cytosol and mitochondria. The assay requires 20 µL of sample for each reaction, which was added to the sample wells. Then a 20 µL Dilution Buffer was added to No SOD and No Xanthine Oxidase (No XO) wells, and a 40 µL Dilution Buffer to blank wells. Thereafter 160 µL of WST working solution was added to samples, controls, standards, and blank wells. The reaction was initiated by the addition of 20 µL xanthine oxidase working solution to the sample and standard well and to the No SOD control wells. The 96-well flat-bottom plate was incubated at 25 °C for 30 min and absorbance was read by using MultiSkan GO microplate spectrophotometer (Thermo Fisher Scientific Inc., Waltham, MA, USA) at 450 nm. The provided Excel-based calculation sheet was used to calculate the test results. Each experiment was repeated in triplicate.

### 2.7. Enzyme-Linked Immunosorbent Assay (ELISA) Measurements

The SH-SY5Y cells were treated the same way as described above. The cell culture media samples were collected after the treatments and stored at −80 °C until the taking of ELISA measurements. TNFα, IL-6, and IL-8 cytokine measurements were determined using human TNFα, IL-6, and IL-8 ELISA assay kits (Invitrogen, Thermo Fisher Scientific Inc., Waltham, MA, USA) according to the manufacturer’s instructions. The absorbance measurements were performed by using a MultiSkan GO microplate spectrophotometer (Thermo Fisher Scientific Inc., Waltham, MA, USA) at 450 nm wavelength. The intensity of the signal is directly proportional to the concentration of the target present in the original samples. The concentrations of cytokines were expressed as pg/mL.

### 2.8. Real-Time PCR

The SH-SY5Y cells were treated evenly as described earlier, in 6-well cell culture dishes (3 × 10^5^ cells/well). After the incubation period, SH-SY5Y cells were washed with PBS and then collected by trypsinization. Total RNA was isolated from all samples using Aurum™ Total RNA Mini Kit (Bio-Rad Inc., Hercules, CA, USA) according to the manufacturer’s instructions. Next, 200 ng of total RNA was used for complementary DNA synthesis performed by using iScript™ Select cDNA Synthesis Kit (Bio-Rad Inc., Hercules, CA, USA) accordingly to the manufacturer’s protocol. The gene expression determinations were carried out in a CFX96 Real-Time System (Bio-Rad Inc., Hercules, CA, USA) using iTaq™ Universal SYBR^®^ Green Supermix (Bio-Rad Inc., Hercules, CA, USA) for the reactions in 20 µL of total reaction volume. Melting curves were generated after each quantitative PCR run to grant that a single specific product was amplified. Bio-Rad CFX Maestro 1.1 software (Bio-Rad Inc., Hercules, CA, USA) was used for the relative quantification by the Livak (∆∆Ct) method. The expression level of the gene of interest was compared to the level of internal control β-actin in all samples. Then these relative expression rates were compared between the treated and untreated samples. The relative expression of the controls was regarded as 1. The mRNA expression of the treated SH-SY5Y cells was compared to the appropriate controls. The primer sequences used in this study are represented in Table 2.

### 2.9. Western Blot

The SH-SY5Y cells were collected by centrifugation after treatments. The cells were lysed with 150 µL of ice-cold lysis buffer (50 mM Tris(hydroxymethyl)aminomethane hydrochloride (Tris-HCl), pH 7.4, 150 mM sodium chloride (NaCl), 0.5% Triton-X 100) containing complete mini protease inhibitor cocktail (Roche Ltd., Basel, Switzerland). The protein concentration of the samples was determined using the Detergent Compatible Protein Assay Kit (Bio-Rad Laboratories, Hercules, CA, USA) and an equal amount of protein was separated from each sample in 10% or 14% polyacrylamide gels. Mini-PROTEAN Tetra Cell polyacrylamide gel electrophoresis system (Bio-Rad Laboratories, Hercules, CA, USA) was used for the vertical electrophoresis. The protein transfer from the gels was carried out by semi-dry electroblotting to nitrocellulose membranes (Pall AG, Basel, Switzerland). A non-fat dry milk (5% (*w*/*v*)) blocking solution (Bio-Rad Laboratories., Hercules, CA, USA) was used for blocking the membranes for 1 h at 25 °C with gentle shaking. Then, the membranes were incubated for 1 h at 25 °C with anti-ferroportin (FPN) IgG (1:1000; Bio-Techne, Minneapolis, MN, USA) and anti-transferrin receptor 1 (TfR1) IgG (1:1000; Thermo Fisher Scientific Inc., Waltham, MA, USA) polyclonal rabbit antibodies. The anti-ferritin heavy chain (FTH) IgG (1:1000; Cell Signaling Technology Europe, Leiden, The Netherlands) polyclonal rabbit antibody was overnight incubated at 4 °C. As a loading control, glyceraldehyde 3-phosphate dehydrogenase (GAPDH) (anti-GAPDH IgG, 1:3000; Merck Life Science Kft., Budapest, Hungary) was used for the Western blots. Horseradish peroxidase (HRP)-linked goat anti-rabbit IgG was used (1:3000; Bio-Rad Laboratories, Hercules, CA, USA) as the secondary antibody, incubated for 1 h at 25 °C. The proteins were detected by using WesternBright ECL chemiluminescent substrate (Advansta Inc., San Jose, CA, USA). The development of the membranes was performed by using the Alliance Q9 Advanced gel documentation system (UVITEC, Cambridge, UK). The optical density of the protein bands was determined by ImageJ software [31] and expressed as a percentage of the target protein/GAPDH ratio.

### 2.10. Measurements of Total Iron

The determination of total iron levels of SH-SY5Y cells was performed by a ferrozine-based colorimetric assay, which was described by Riemer et al. [32]. Briefly, after 24 h of the resting period, 1 × 10^6^ SH-SY5Y cells were treated on 25 cm^2^ cell culture flasks and the cells were collected after the incubation. For the lysis, 50 mM sodium hydroxide (NaOH) was used at 25 °C for 2 h. Then the samples were mixed with an iron-releasing reagent containing 1.4 M hydrogen chloride (HCl) (4.5% (wt/vol)) and potassium permanganate (KMnO_4_) in distilled water (dH_2_O) and were incubated at 60 °C for 2 h. Afterward, the samples were cooled to 25 °C and for chelating iron, an iron detection reagent containing 6.5 mM ferrozine, 6.5 mM neocuproine, 2.5 M ammonium acetate, and 1 M ascorbic acid was added to each sample and incubated at 25 °C for 30 min. The absorbance was detected by using MultiSkan GO spectrophotometer (Thermo Fisher Scientific Inc., Waltham, MA, USA) at 550 nm wavelength. The iron concentration was calculated by a ferric chloride (FeCl_3_) (0–300 µM) standard curve treated as evenly as the samples. A DC Protein Assay Kit (Bio-Rad Inc., Hercules, CA, USA) was used to measure the protein concentration of each sample. The iron level was normalised against the protein concentration and was expressed as µM iron/mg protein.

### 2.11. Measurements of Heme Concentration

An aqueous alkaline solution method-based Heme assay kit (Sigma-Aldrich Kft., Budapest, Hungary) was used for the heme detection. For the measurements, 3 × 10^5^ SH-SY5Y cells were treated as previously described. The method was carried out according to the manufacturer’s instructions. Briefly, SH-SY5Y cells were collected after the treatments and lysed in 50 µL of dH_2_O. The same amount of dH_2_O was served as blank. The experiments were carried out on 96-well plates. For the calibration controls, 50 µL of Heme Calibrator and 50 µL of dH_2_O were added into the wells, followed by the addition of 200 µL of dH_2_O into the blank and the Heme Calibrator reaction wells. Next, 50 µL of each sample, then 200 µL of Heme Reagent was added into sample wells. The plate was incubated for 5 min at 25 °C. The experiments were carried out in triplicate in the four independent experiments. The diluted Heme Calibrator refers to 62.5 M heme. The converted heme form produces a colorimetric result, of which absorbance was measured by using MultiSkan GO microplate spectrophotometer (Thermo Fisher Scientific Inc., Waltham, MA, USA) at 400 nm wavelength, where the OD is directly proportional to the heme concentration in the sample. The concentration of heme was expressed as µM.

### 2.12. Statistical Analysis

In this work, the presented data are representative of at least three independent experiments. For all data, n corresponds to the number of independent experiments. CAT activity assay, SOD assay, Real-time PCR reactions, ELISA assay kits, total iron level measurements, and heme measurements were performed in triplicate in all independent experiments. SH-SY5Y cell viability assays and ROS detections were measured in quadruplicate in each independent experiment. SPSS software (IBM Corporation, Armonk, NY, USA) was used for the statistical analysis. The statistical significance was determined using ANOVA analyses with Tukey HSD post-hoc tests to compare lutein and Glu-treated groups (24 h, 48 h, and 72 h) to their control group (24 h, 48 h, and 72 h) and to determine the significant difference between the treated groups (e.g., lutein vs. Glu with lutein). Data are presented as mean values with error bars corresponding to the standard deviation (SD). The statistical significance was set at *p*-value < 0.05 in each analysis. The * indicates *p* < 0.05 between Glu treatment and control cells, # indicates *p* < 0.05 between Glu treatment and lutein with Glu treatment, ‡ indicates *p* < 0.05 between lutein treatment and control cells.

## 3. Results

### 3.1. Effects of Lutein on Reactive Oxygen Species (ROS) Generated by Glutamate

To examine the ROS-generating effect of the excitatory neurotransmitter Glu, we used two concentrations of Glu, 1 mM, and 5 mM in our experiments, which were not apoptosis inducers according to our concentration and time-dependent preliminary viability experiments (Appendix A). We observed that 1 mM and 5 mM Glu triggered ROS production in SH-SY5Y cells, and we determined ROS after the addition of Glu in 10 to 30 min. We examined the possible effect of the antioxidant lutein on Glu-induced ROS production in the cells; 10 ng/µL lutein alone did not attenuate ROS in the SH-SY5Y cells. Both concentrations of Glu significantly induced the intracellular ROS compared to the controls. The addition of lutein with Glu together significantly decreased Glu-induced ROS compared to the Glu treatments (Figure 2).

### 3.2. Effects of Lutein on Antioxidant Enzyme Activities of Glutamate-Treated SH-SY5Y

To check the defence mechanisms against Glu-induced oxidative stress, first CAT enzyme activity measurements were performed in SH-SY5Y cells (Figure 3A). Both 1 mM and 5 mM Glu significantly induced the CAT enzyme activity in the SH-SY5Y cells after 24 h, 48 h, and 72 h. In the mutual (lutein with Glu) treatment the presence of 10 ng/µL of lutein significantly decreased the CAT enzyme activity in the cells.

Next, we examined the SOD enzyme activity after the Glu and lutein treatments. Lutein significantly elevated SOD activity in the cells after 24 h and 48 h with a decreasing tendency. SOD activity was significantly decreased at 48 h and 72 h after both concentrations of Glu treatments in SH-SY5Y cells. In the case of lutein treatment with Glu together, SOD activity was increased compared to Glu treatments (Figure 3B).

### 3.3. Effects of Lutein on Proinflammatory Cytokine Secretion Induced by Glutamate in SH-SY5Y Cells

To investigate the effects of 1 mM and 5 mM Glu on the inflammation in SH-SY5Y cells, IL-6, IL-8, and TNFα proinflammatory cytokine ELISA measurements were performed (Figure 4A–C). Culturing the cells with 10 ng/µL of lutein significantly decreased the TNFα secretion after 24 h compared to controls. The addition of 1 mM and 5 mM Glu elevated the TNFα cytokine secretion after 24 h, 48 h, and 72 h compared to controls. The use of lutein with Glu together resulted in significantly reduced TNFα levels compared to the Glu treatments (Figure 4A).

Next, IL-6 cytokine levels were performed by ELISA. The proinflammatory IL-6 levels were slightly but significantly increased after 1 mM and 5 mM Glu treatments at 24 h, 48 h, and 72 h. Treatment with 10 ng/µL of lutein significantly decreased the IL-6 secretion in SH-SY5Y cells at 48 h. Lutein in the cultures with 1 mM and 5 mM Glu reduced the IL-6 levels after each treatment (Figure 4B).

We also examined the secreted level of the chemoattractant IL-8. Treatment with 10 ng/µL of lutein alone did not modulate the measured IL-8 in SH-SY5Y cells. Adding 1 mM and 5 mM Glu to the cells, significantly elevated the IL-8 levels at 24 h, 48 h, and 72 h, as well. In the mutual (lutein with Glu) culturing the presence of lutein decreased the IL-8 levels in the cells (Figure 4C).

According to our ELISA measurements, all three examined proinflammatory cytokine secretion was increased upon Glu treatments. Meanwhile, in the presence of lutein with Glu, lutein showed a downregulatory effect on the inflammatory cytokine levels in SH-SY5Y cells.

### 3.4. Lutein Ameliorates the Effect of Glutamate on the mRNA Expression of the Iron Regulatory Hormone Hepcidin

The inflammatory response is tightly bound to the iron levels and its homeostasis [33], thereby we were interested in the modulatory effect of Glu on iron metabolism. First, we investigated the hepcidin coding HAMP gene expression measurements because the iron-regulatory hormone hepcidin is known to elevate during inflammation [34]. The relative mRNA expression of HAMP showed a rising trend after 5 mM Glu treatment. Lutein cultured with Glu together reduced the expression of HAMP in the cells. Interestingly, 10 ng/µL of lutein with 1 mM Glu significantly decreased the level of HAMP only at 24 h. Meanwhile, treatment using lutein with 5 mM Glu significantly downregulated the expression of HAMP in each treatment (Figure 5).

### 3.5. Lutein Modifies the Glutamate’s Effect on Iron Uptake and Release

According to changes in the HAMP expression, we assumed the connection between the effect of Glu and iron homeostasis on SH-SY5Y cells. Therefore, we continued to examine the alterations in iron uptake, release, and storage.

We examined the relative mRNA expression of divalent metal transporter 1 (DMT1) and transferrin receptor 1 (TfR1) to determine alteration in iron uptake in addition to the only known iron exporter ferroportin (FPN) expression to follow the releasing iron upon Glu and lutein treatments at 24 h, 48 h, and 72 h.

The relative mRNA expression of TfR1 increased after 1 mM and 5 mM Glu treatments at 24 h. The presence of lutein significantly decreased the TfR1 mRNA expression compared to 1 mM and 5 mM Glu (Figure 6A). TfR1 protein levels decreased significantly upon 1 mM and 5 mM Glu treatments, and lutein in the presence of 1 mM Glu elevated the TfR1 protein levels compared to Glu treatment (Figure 6B,C). The mRNA expression of DMT1 showed no significant change at 24 h, and the FPN levels remained consistent, as well (Figure 6A). The protein expression of FPN showed no alteration at 24 h (Figure 6B,D).

Upon 48 h treatments, the relative mRNA expression of TfR1 increased after 5 mM Glu treatment, while the presence of lutein decreased the expression of TfR1 in SH-SY5Y cells (Figure 7A). According to the protein analyses, the expression of TfR1 decreased in the lutein and lutein with 1 mM Glu-treated SH-SY5Y cells (Figure 7B,C). Relative mRNA expression of FPN elevated after 1 mM Glu treatment, while lutein with Glu significantly decreased the expression compared to Glu treatments alone (Figure 7A). The protein expression of FPN showed elevation in the lutein with 1 mM Glu-treated cells at 48 h (Figure 7B,D). In our experiments, DMT1 showed no change in the relative mRNA expression at 48 h (Figure 7A).

Based on our results of the 72 h measurements, both 1 mM and 5 mM Glu concentrations significantly increased the relative mRNA expression of TfR1, while lutein alone and in culturing together with Glu downregulated the expression of TfR1 (Figure 8A). Protein expression of TfR1 was downregulated in the lutein-treated cells and reduced further when Glu was cultured together with lutein. Indeed, 1 mM Glu treatment increased the TfR1 protein expression, while 5 mM Glu decreased it. (Figure 8B,C). The FPN relative mRNA expression remained unchanged in the 72 h experiments (Figure 8A). The protein analyses of FPN showed a slight decrease when lutein was added to the cells together with 1 mM or 5 mM Glu (Figure 8B,D). The relative mRNA expression of DMT1 elevated after 5 mM Glu treatment and decreased when lutein was also present in the media (Figure 8A).

### 3.6. Effects of Glutamate and Lutein on the Iron Storage of the SH-SY5Y Cells

Henceforward, to identify the changes in the iron storage, the mRNA expression levels of the major iron storage protein ferritin heavy chain (FTH) and mitochondrial ferritin (FTMT) were determined after the treatments.

The relative mRNA expression of FTH showed an upregulation in the cells treated with lutein and Glu together at 24 h (Figure 9A). The protein expression showed elevation upon lutein with 1 mM Glu treatment (Figure 9B,C). The relative mRNA expression of FTMT was raised in the 1 mM and 5 mM Glu-treated cells and decreased when lutein was also present in the Glu-treated cultures (Figure 9A).

The FTH relative mRNA expression was decreased in the cells which were treated with 1 mM, 5 mM Glu, and lutein together (Figure 10A). The protein expression increased in the 1 mM Glu and lutein-treated SH-SY5Y cells (Figure 10B,C). The relative mRNA expression of FTMT was elevated after 1 mM and 5 mM Glu treatment at 48 h (Figure 10A).

In the 72 h treatments, a decrease was detected in the relative mRNA expression of FTH in the 1 mM Glu-treated cells, and in the presence of lutein with 5 mM Glu in the cultures (Figure 11A). According to the protein analyses, the expression of FTH was downregulated in the 1 mM Glu-treated cells and in the 5 mM Glu together with lutein-treated cells compared to the 5 mM Glu treatment (Figure 11B,C). The mRNA expression of FTMT was upregulated upon Glu treatments and decreased when lutein was cultured together with Glu (Figure 11A).

Based on the results, lutein downregulated the iron uptake in SH-SY5Y cells and decreased Glu-induced iron uptake.

### 3.7. Effect of Lutein on the Iron Levels in the Glutamate-Treated SH-SY5Y Cells

Since Glu and lutein treatment modified the iron storage and uptake in SH-SY5Y cells, the next total iron measurements were performed to follow the iron status of the cells. Both concentrations of Glu showed an upward trend in the total iron content, which significantly increased at 24 h, 48 h, 72 h compared to controls (Figure 12A). Lutein alone did not change the total iron levels of the SH-SY5Y cells. Lutein with 1 mM and 5 mM Glu together significantly decreased the total iron levels to a similar value as the initial iron content of the cells. Upon total iron measurements, lutein seems to prevent the Glu-induced accumulation of iron in the cells as a protection against excessive iron-related ROS production.

Next, heme measurements were performed for a better understanding of the modifications in the iron levels. Heme showed a significant upregulation upon lutein treatments with a peak at 48 h. Culturing SH-SY5Y cells with 1 mM and 5 mM Glu resulted in a reduced heme at both 24 h, 48 h, and 72 h compared to controls. Lutein with Glu together appeared in similar alteration as lutein alone, significantly increased heme concentration after each treatment compared to Glu treatments, with a maximum at 48 h (Figure 12B).

### 3.8. Lutein Downregulates the Glutamate-Induced Upregulation of Lipoxygenases

Oxidative stress induced by ROS can trigger lipid peroxidation, which interferes with neuronal activity [35]. Therefore, we carried out gene expression measurements on lipoxygenases implicated in lipid peroxidation.

ACSL4, ALOX5, and ALOX15 mRNA expressions were determined at 24 h, 48 h, and 72 h (Figure 13A–C).

The relative mRNA expression of ALOX5 and ALOX15 was significantly upregulated at 24 h, while the presence of lutein decreased the ALOX5 expression in SH-SY5Y cells. The expression of ACSL4 increased upon 5 mM Glu treatment, while its expression decreased when cultured together with lutein (Figure 13A).

The relative mRNA expression of ALOX5 further increased in the Glu-treated cells at 48 h, and the addition of lutein to 5 mM Glu significantly downregulated the expression. ALOX15 expression showed an elevation upon 5 mM Glu treatment while downregulated in the presence of lutein. The expression of ACSL4 was only increased after 5 mM Glu treatment and decreased together with lutein (Figure 13B).

The expression of ALOX5 remained upregulated after being treated with Glu at 72 h. Meanwhile, the addition of lutein to the cultures reduced the expression. Indeed, the relative expression of both ALOX15 and ACSL4 was upregulated in the 5 mM Glu-treated SH-SY5Y cells and decreased when lutein was added to the media (Figure 13C).

Based on these findings, Glu seems to induce ROS and inflammation, downregulate SOD enzyme activity, provoke iron accumulation and possibly initiate lipid peroxidation in SH-SY5Y cells. Meanwhile, lutein decreased the Glu-induced ROS, downregulated proinflammatory cytokine secretion, prevented the cells from iron accumulation, and decreased the expression of the examined gene related to lipid peroxidation.

## 4. Discussion

The xanthophyll carotenoid lutein has been widely used as supplementation according to its antioxidant, and anti-inflammatory properties, thus mainly according to its protective effects against light-induced oxidative damage, to prevent or decrease the risk for development of age-related macular degeneration and cataracts [2,3]. Oxidative stress is related to neurodegeneration, with an injurious effect on the progression of neurodegenerative diseases including Alzheimer’s disease, Parkinson’s disease, and Amyotrophic Lateral Sclerosis [36,37]. Previously lutein has been reported by our research group to upregulate antioxidant enzymes, elevate the anti-inflammatory cytokine secretion, and suppress ROS in H_2_O_2_-induced conditions in BV-2 murine microglia [5].

Glu is the main and most abundant excitatory neurotransmitter in the CNS, playing an important role in physiological brain functioning, modelling memory, and learning [7]. Excessive Glu provokes neuronal dysfunction and degeneration, thus Glu toxicity is a possible way to model neurodegenerative diseases [38,39,40]. According to the literature, 8 mM to 80 mM Glu is excitotoxic to SH-SY5Y cells through the hyperactivation of Glu receptors [41,42,43]. In our studies, two concentrations of Glu were chosen, which were not cytotoxic to SH-SY5Y cells but assumed to be possible to evoke the process of neuronal dysregulation.

Oxidative stress plays a role in several metabolic and neurodegenerative disorders and is also one of the leading mechanisms of Glu-induced neurotoxicity [36,44,45]. Glu has been reported to induce intracellular ROS generation in hippocampal neuronal cells; in addition, recently it has been published that an excess amount of Glu increases the mitochondrial ROS in SH-SY5Y cells [46,47]. In our experiments, a significantly increased ROS generation was detected upon Glu treatment in SH-SY5Y cells. We determined the effect of lutein on ROS generation in Glu-treated cells. The presence of lutein alone did not modify ROS levels while lutein together with Glu resulted in a significant reduction of ROS suggesting an amending effect of lutein on ROS production.

CAT is one of the major antioxidant enzymes. Its dysregulation is related to the imbalance of ROS regulation and the pathogenesis of age-related diseases including neurodegenerative disorders [48]. CAT has been reported with a protective effect against Glu-induced neurotoxicity in rat hippocampal neurons [49]. In our experiments, Glu induced the CAT enzyme activity in SH-SY5Y, while the presence of lutein decreased the CAT activity, suggesting a state of decreased oxidative insult due to antioxidant lutein in SH-SY5Y cells.

SOD enzymes are important members of the antioxidant defence mechanisms against oxidative stress in cells. Misfunctioning in SOD due to coding gene mutations are identified in motor neuron diseases [50,51,52]. SOD enzyme activity has been reported to decrease upon Glu treatment, which is in correlation with our results [41]. SOD activity has been published to elevate upon antioxidant lutein treatment in ARPE19 retinal pigment epithelia, in BV-2 murine microglia and uveitis [5,53,54]. In our experiments, lutein increased the SOD activity and compensated for the downregulatory effect of Glu in SH-SY5Y cells.

Neuroinflammation is a response that can be induced by diseases, stress, infection, or neuronal damage. The development is mediated by cytokines including TNFα, IL-6, IL-1β, IFNγ, and IL-8, among chemokines, ROS, and secondary messenger molecules [14,15,16].

TNFα was demonstrated with an additive effect of Glu neurotoxicity in human fetal brain cell cultures during neuronal injury [55]. In our results, Glu significantly increased the TNFα secretion of SH-SY5Y cells.

Controversial effects have been published of IL-6. The chronic levels of IL-6 have been reported to disturb neuronal function via activation of metabotropic Glu receptors in Purkinje neurons, increased levels have been demonstrated in inflammatory disorders in CNS. In contrast, IL-6 can exert neuronal survival in controlled conditions [56,57,58]. In our measurements, the IL-6 secretion was slightly but significantly increased in Glu treatment.

IL-8 has been demonstrated to increase cerebrospinal fluid in neuroinflammation [16]. Upon our Glu treatment, a significant elevation was detected in the secretion of IL-8.

Lutein treatment has been reported to downregulate the expression of TNFα, IL-6, and IL-1β in inflammation and oxidative stress conditions in BV-2 microglia [5,59].

In our experiments, lutein significantly reduced the elevated TNFα, IL-6, and IL-8 secretion upon Glu treatments. Decreasing the proinflammatory cytokine secretions is a pivotal part of the protective mechanisms in inflammation.

The inflammatory response is tightly bound to the iron levels and its homeostasis [33,60]. The major regulatory hormone hepcidin has been reported to maintain brain homeostasis. It has been published to upregulate inflammation and neuroinflammation [61,62,63]. In our study, we were interested in the modulatory effect of Glu and lutein on iron metabolism. Hepcidin coding HAMP showed a significant elevation in the relative mRNA expression in Glu treatments of SH-SY5Y cells. The presence of lutein decreased the expression of HAMP in our treatments.

Iron is an essential element for the proper functioning of the brain; meanwhile, the regulation in uncontrolled conditions or excess levels can be harmful to the cells [64]. To reveal the alterations in iron metabolism, we performed gene expression measurements of the genes responsible for iron uptake, release, and storage.

TfR1 and DMT1 expression was reported to upregulate in THA-induced Glu neurotoxicity in rat spinal cord motor neurons while FTH decreased [65]. Our results show that TfR1 is involved whereas DMT1 does not in the iron uptake of Glu-treated SH-SY5Y cells. According to the results, Glu response promotes the uptake of iron through the upregulation of TfR1. In our data, Glu upregulated the gene expression of TfR1 at 24 h and 72 h, while the protein expression was only elevated at 72 h, suggesting a fluctuation in the mRNA synthesis and a delay between mRNA and protein expressions. Lutein with 1 mM Glu downregulated the TfR1 protein expression upon all three duration treatments, but only at 72 h with 5 mM Glu. These results suppose that lutein may decrease Glu-mediated iron uptake via TfR1.

TfR1 is known to be upregulated during ferroptosis [66]. In our experiments, TfR1 showed an increase in the relative mRNA expression which was followed by protein expression at 72 h. Due to its high ferroxidase activity, FTH is an important negative regulator of ferroptosis [67,68]. Moreover, the downregulation of FPN contributes to iron retention in the cells and triggers ferroptosis [69]. According to these facts, we monitored FTH and FPN mRNA and protein levels.

The iron efflux by FPN showed no alterations at protein levels upon Glu treatment suggesting that the iron export was not inhibited in SH-SY5Y cells. Interestingly, lutein decreased the FPN levels in 1 mM and 5 mM Glu treatments, but only at 72 h.

In our experiments, Glu did not change or decrease the FTH levels. Interestingly, lutein together with 1 mM Glu increased the FTH levels, while it decreased the expression in 5 mM Glu-treated SH-SY5Y cells at 72 h.

Since the mRNA levels of FTH decreased upon Glu treatment, the mRNA expression of the mitochondrial iron storage protein was also determined to reveal whether Glu modifies the mitochondrial iron metabolism. In our experiments, FTMT expression showed significant upregulation upon Glu treatments, suggesting that the iron trafficking into mitochondria.

To clarify the effect of Glu and lutein on iron levels, total iron measurements and heme measurements were carried out. Total intracellular iron levels were elevated in the Glu treatments, showing an increasing tendency over time. The heme levels decreased upon Glu treatment. On the contrary, lutein reduced the Glu-induced increase in total iron content, while upregulated heme in SH-SY5Y cells.

According to the results, the total iron levels and iron uptake are increased by Glu treatment, but the FTH protein level does not show upregulation. On the other hand, the FTMT mRNA expression is strongly increased upon Glu treatment. These data presuppose the alterations of mitochondrial iron utilization, which is possibly caused by the intracellular iron trafficking from the cytosol to the mitochondria and/or by the changes in iron uptake and release.

According to the literature, iron accumulation is also able to trigger ferroptosis and preponderant iron-dependent lipid peroxidation [70].

Overexpression of lipid-peroxidising ALOX5 and ALOX15 is related to several diseases such as diabetes, systemic sclerosis, atherosclerosis, prostate cancer, and colorectal cancer [27,28], besides, mediating proinflammatory lipid peroxidation and the main promoter of ferroptosis by producing lipid hydroperoxides [71,72]. Recently it has been published that ACSL4 participates in lipid peroxide generation in a manner through its phosphorylation, which amplifies lipid peroxidation [73].

Our results assumed that Glu upregulated both ALOX5, ALOX15, and ACSL4 expression, meanwhile, lutein decreased the expression of the examined genes suggesting the protecting role of lutein against lipid peroxidation and neuronal cell death.

In our experiments, Glu increased the TfR1 protein level suggesting an enhanced iron uptake into the cells. Moreover, Glu elevated the total intracellular iron content over time, while the presence of lutein reduced the iron level back to control. In heme measurements, a decrease could be detected in heme level after Glu treatment, presupposing that Glu alters mitochondrial functions, which were upregulated by lutein treatments. These changes may indicate the development of ferroptosis. Based on our findings, Glu activated lipid peroxidation by increasing the expression of genes involved in lipid peroxidation, which was downregulated by lutein.

## 5. Conclusions

According to our results, lutein provided protection for SH-SY5Y neuroblastoma cells against Glu-induced oxidative stress and proinflammatory cytokine production as well as attenuated the iron accumulation and the expression of the lipid-peroxidising genes.

Based on the observations, lutein could be a beneficial supportive treatment in neurodegenerative disorders.

## Figures and Tables

**Figure 1 antioxidants-11-02269-f001:**
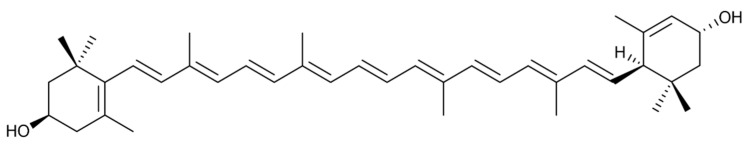
The structure of (all-*E*)-Lutein ((all-*E*,3*R*,3′*R*,6′*R*)-3,3′-Dihydroxy-β,ε-carotene).

**Figure 2 antioxidants-11-02269-f002:**
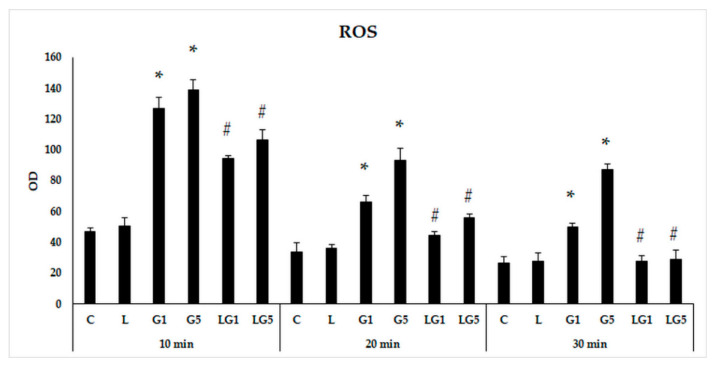
Determination of the effect of lutein on Reactive Oxygen Species (ROS) generated by Glu. SH-SY5Y cells were treated with 10 ng/µL of lutein, 1 mM or 5 mM Glu, or lutein with Glu together to detect the changes in ROS. The cells were incubated for 10, 20, and 30 min at 37 °C and 5% CO_2_. After incubation, the cells were stained with ROS deep red dye solution and incubated for 30 min at 37 °C and 5% CO_2_. The ROS production was determined by a Fluorometric Intracellular ROS Kit. The ROS changes were determined as a percentage of control. The bars represent the mean values (±standard deviation error bars, SD) of three independent experiments (*n* = 3), each performed in quadruplicate, and are presented relative to their own control cells. For better transparency, the measured ROS of DMSO controls are not presented, each OD of controls detected between zero to 50. The * marks the statistical significance of Glu treatments compared to control, # shows the statistical significance of combined (lutein with Glu) treatments compared to the Glu treatments at 10 min, 20 min, or 30 min (*p* < 0.05). Abbreviations of treatments: L-10 ng/µL of lutein; C-absolute control; G1-1 mM Glu, G5-5 mM Glu, LG1-10 ng/µL lutein with 1 mM Glu; LG5-10 ng/µL lutein with 5 mM Glu, OD-optical density.

**Figure 3 antioxidants-11-02269-f003:**
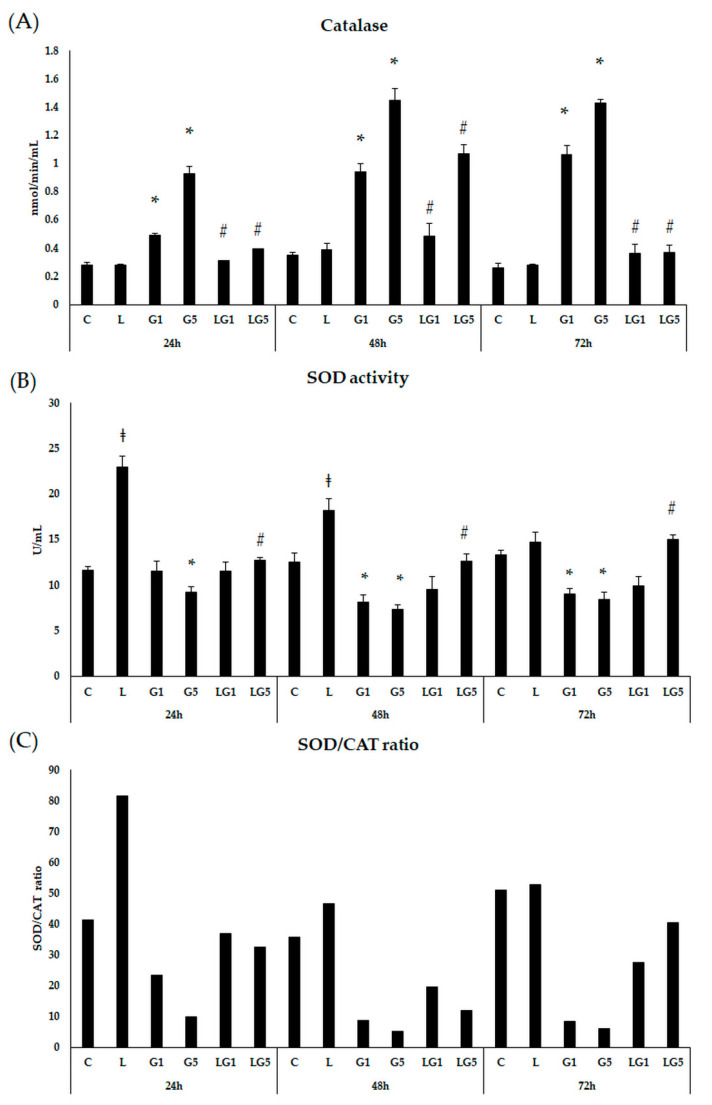
Determination of the effects of lutein on CAT activity (**A**), SOD activity (**B**), and SOD/CAT ratio (**C**) of the SH-SY5Y cells. SH-SY5Y cells were treated with 10 ng/µL of lutein, 1 mM, or 5 mM Glu, or lutein with Glu together to detect the changes in enzyme activity. The cells were incubated for 24 h, 48 h, and 72 h at 37 °C and 5% CO_2_. After incubation, the activity of the CAT enzyme was determined by the Catalase Assay kit, and the result was expressed as nmol/min/mL. The activity of the SOD enzyme was determined by the Superoxide Dismutase Activity Assay kit, and the result was expressed as U/mL. The bars represent the mean values (±standard deviation error bars, SD) of three independent experiments (*n* = 3) performed in triplicate and are presented relative to their control cells. The * marks the statistical significance of Glu treatments compared to control, ‡ shows the statistical significance of lutein treatments compared to control, # signs the statistical significance of combined (lutein with Glu) treatments compared to the Glu treatments at 24 h, 48 h, and 72 h (*p* < 0.05).

**Figure 4 antioxidants-11-02269-f004:**
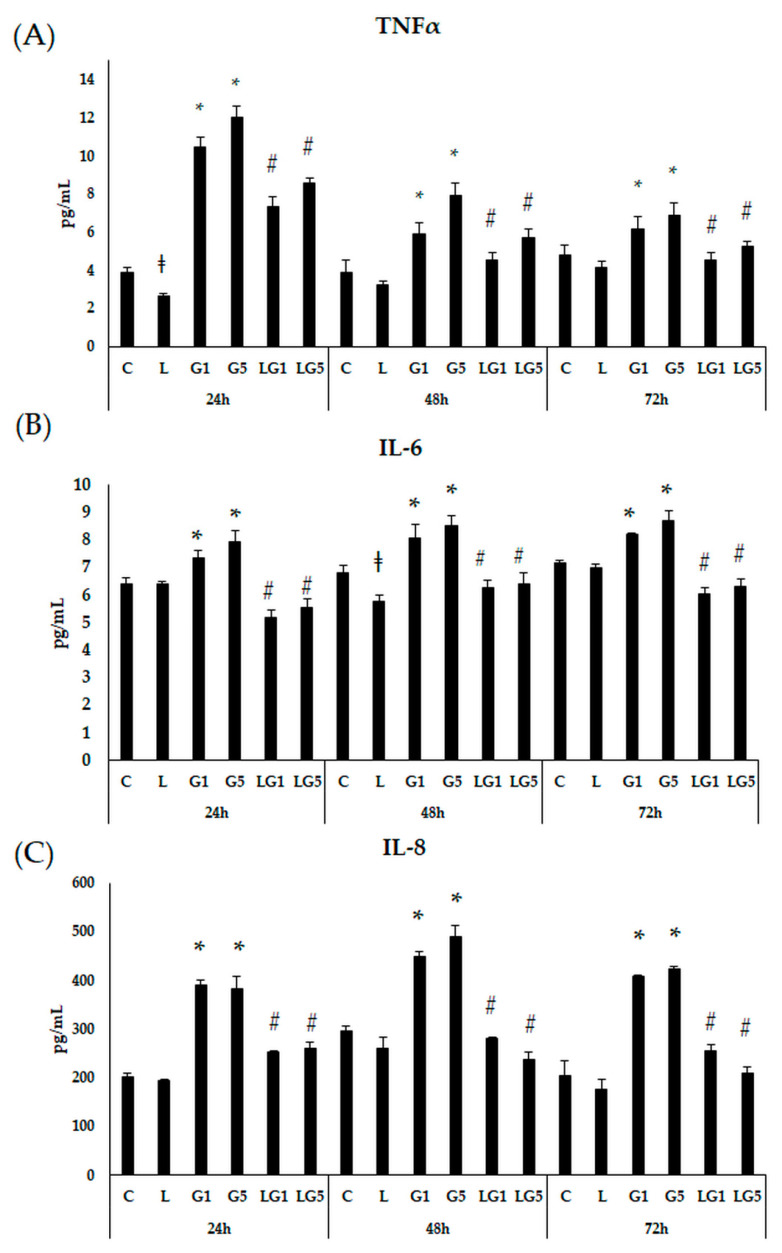
Determination of proinflammatory TNFα (**A**), IL-6 (**B**), and IL-8 (**C**) secretion after lutein and Glu treatments of the SH-SY5Y cells. SH-SY5Y cells were treated with 10 ng/µL of lutein, 1 mM, 5 mM Glu, or lutein with Glu together to detect the changes in the secretion. The cells were incubated for 24 h, 48 h, and 72 h at 37 °C and 5% CO_2_. After incubation, the cell culture media was used to determine the cytokine secretion by human TNFα ELISA kit (**A**), by human IL-6 ELISA kit (**B**) by human IL-8 ELISA kit (**C**), and the result was expressed as pg/mL. The bars represent the mean values (±standard deviation error bars, SD) of three independent experiments (*n* = 3) performed in triplicate and are presented relative to their control cells. The * marks the statistical significance of Glu treatments compared to control, ‡ shows the statistical significance of lutein treatments compared to control, # signs the statistical significance of combined (lutein with Glu) treatments compared to the Glu treatments at 24 h, 48 h, and 72 h (*p* < 0.05).

**Figure 5 antioxidants-11-02269-f005:**
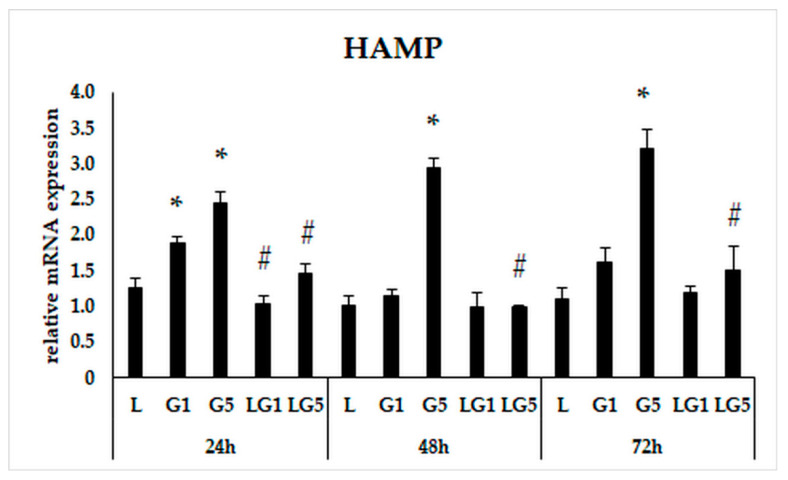
Determination of mRNA expression of HAMP after lutein and Glu treatments of the SH-SY5Y cells. SH-SY5Y cells were treated with 10 ng/µL of lutein, 1 mM, 5 mM Glu, or lutein with Glu together to detect the changes in the expression of HAMP. The gene expression measurements were performed by SYBR Green RT-PCR protocol using β-actin as a housekeeping gene. The relative expression of controls was regarded as 1. The bars represent the mean values (±standard deviation error bars, SD) of three independent experiments (*n* = 3) performed in triplicate and are presented relative to their control cells. The * marks the statistical significance of Glu treatments compared to control, # signs the statistical significance of combined (lutein with Glu) treatments compared to the Glu treatments at 24 h, 48 h, and 72 h (*p* < 0.05).

**Figure 6 antioxidants-11-02269-f006:**
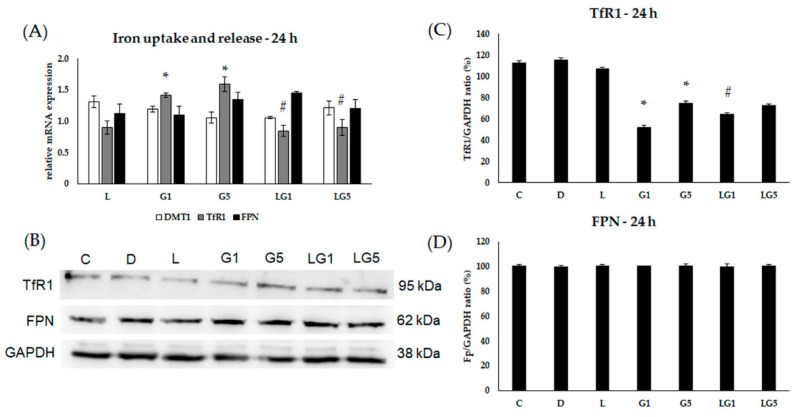
Determination of mRNA expression (**A**) and protein (**B**–**D**) levels of TfR1, DMT1, and FPN after lutein and Glu treatments of the SH-SY5Y cells at 24 h SH-SY5Y cells were treated with 10 ng/µL of lutein, 1 mM, 5 mM Glu, lutein with Glu together to detect the changes in the expression of TfR1, DMT1, and FPN. The gene expression measurements were performed by SYBR Green RT-PCR protocol using β-actin as a housekeeping gene. The relative expression of controls was regarded as 1. The protein expression measurements were performed by semi-dry Western blotting using GAPDH as a loading control. The protein’s OD was expressed as a percentage of the target protein/GAPDH ratio. The bars represent the mean values (±standard deviation error bars, SD) of three independent experiments (*n* = 3) performed in triplicate and are presented relative to their control cells. The * shows the statistical significance of Glu treatments compared to control, # signs the statistical significance of combined (lutein with Glu) treatments compared to the Glu treatments at 24 h (*p* < 0.05).

**Figure 7 antioxidants-11-02269-f007:**
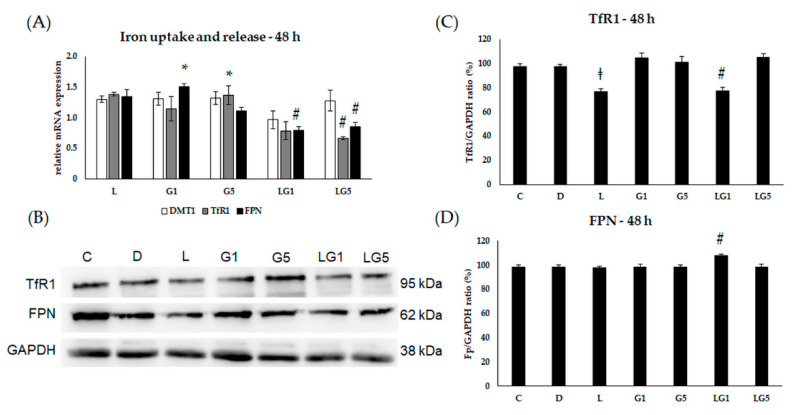
Determination of mRNA expression (**A**) and protein (**B**–**D**) levels of TfR1, DMT1, and FPN after lutein and Glu treatments of the SH-SY5Y cells at 48 h. SH-SY5Y cells were treated with 10 ng/µL of lutein, 1 mM, 5 mM Glu, lutein with Glu together to detect the changes in the expression of TfR1, DMT1, and FPN. The gene expression measurements were performed by SYBR Green RT-PCR protocol using β-actin as a housekeeping gene. The relative expression of controls was regarded as 1. The protein expression measurements were performed by semi-dry Western blotting using GAPDH as a loading control. The protein’s OD was expressed as a percentage of the target protein/GAPDH ratio. The bars represent the mean values (±standard deviation error bars, SD) of three independent experiments (*n* = 3) performed in triplicate and are presented relative to their control cells. The * marks the statistical significance of Glu treatments compared to control, ‡ shows the statistical significance of lutein treatments compared to control, # signs the statistical significance of combined (lutein with Glu) treatments compared to the Glu treatments at 48 h (*p* < 0.05).

**Figure 8 antioxidants-11-02269-f008:**
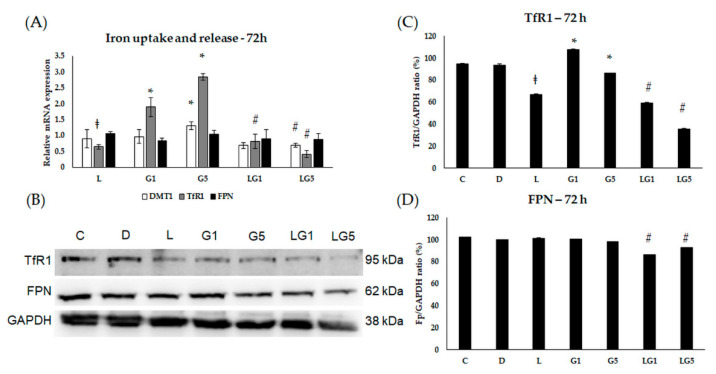
Determination of mRNA expression (**A**) and protein (**B**–**D**) levels of TfR1, DMT1, and FPN after lutein and Glu treatments of the SH-SY5Y cells at 72 h. SH-SY5Y cells were treated with 10 ng/µL of lutein, 1 mM, 5 mM Glu, or lutein with Glu together to detect the changes in the expression of TfR1, DMT1, and FPN. The gene expression measurements were performed by SYBR Green RT-PCR protocol with β-actin as a housekeeping gene. The relative expression of controls was regarded as 1. The protein expression measurements were performed by semi-dry Western blotting using GAPDH as a loading control. The protein’s OD was expressed as a percentage of the target protein/GAPDH ratio. The bars represent the mean values (±standard deviation error bars, SD) of three independent experiments (*n* = 3) performed in triplicate and are presented relative to their control cells. The * marks the statistical significance of Glu treatments compared to control, ‡ shows the statistical significance of lutein treatments compared to control, # signs the statistical significance of combined (lutein with Glu) treatments compared to the Glu treatments at 72 h (*p* < 0.05).

**Figure 9 antioxidants-11-02269-f009:**
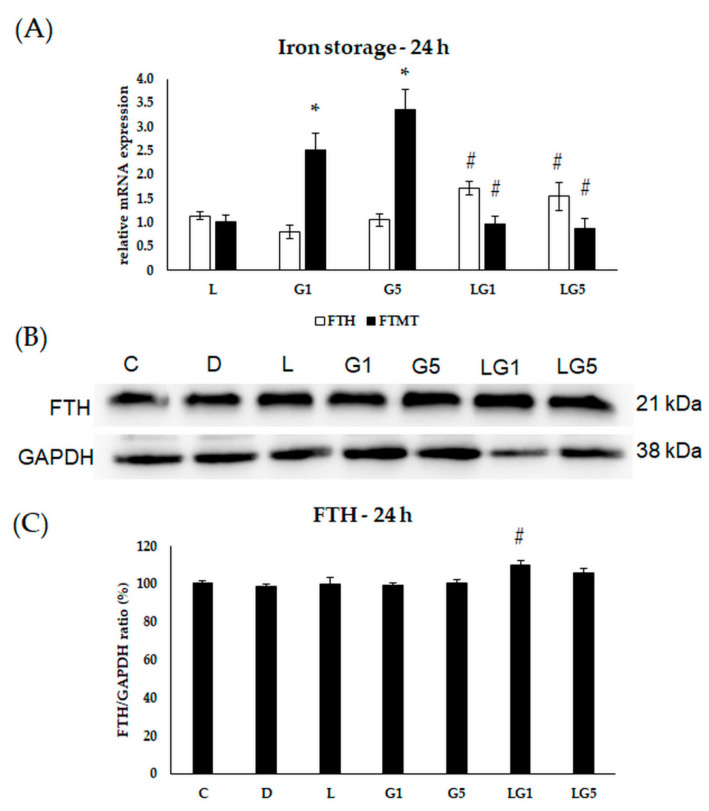
Determination of mRNA expression (**A**) and protein (**B**,**C**) levels of iron storing FTH and FTMT after lutein and Glu treatments of the SH-SY5Y cells at 24 h. SH-SY5Y cells were treated with 10 ng/µL of lutein, 1 mM, 5 mM Glu, or lutein with Glu together to detect the changes in the expression of FTH and FTMT. The gene expression measurements were performed by SYBR Green RT-PCR protocol using β-actin as a housekeeping gene, the relative expression of controls was regarded as 1. The protein expression measurements were performed by semi-dry Western blotting using GAPDH as a loading control. The protein’s OD was expressed as a percentage of the target protein/GAPDH ratio. The bars represent the mean values (±standard deviation error bars, SD) of three independent experiments (*n* = 3) performed in triplicate and are presented relative to their control cells. The * marks the statistical significance of Glu treatments compared to control, # shows the statistical significance of combined (lutein with Glu) treatments compared to the Glu treatments at 24 h (*p* < 0.05).

**Figure 10 antioxidants-11-02269-f010:**
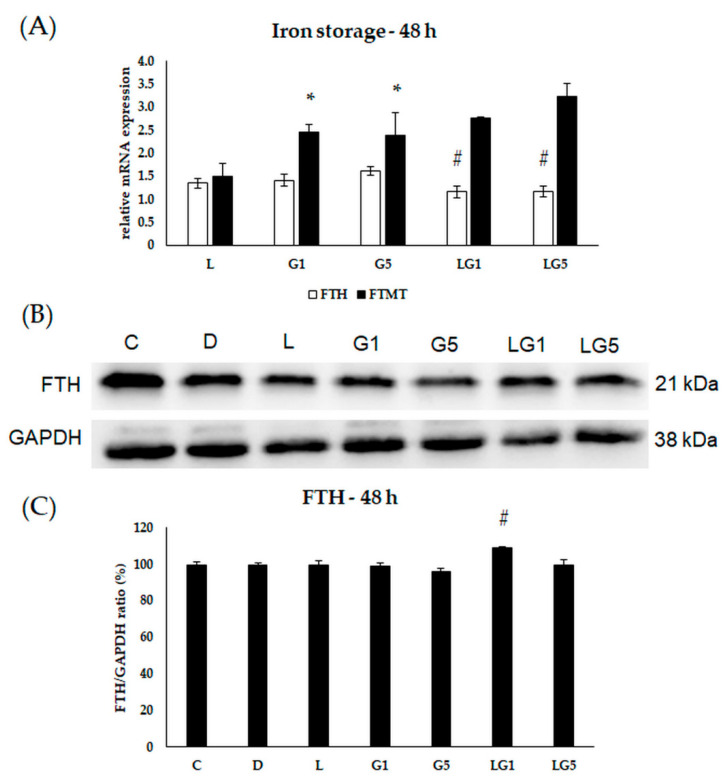
Determination of mRNA expression (**A**) and protein (**B**,**C**) levels of iron storing FTH and FTMT after lutein and Glu treatments of the SH-SY5Y cells at 48 h. SH-SY5Y cells were treated with 10 ng/µL of lutein, 1 mM, 5 mM Glu, or lutein with Glu together to detect the changes in the expression of FTH and FTMT. The gene expression measurements were performed by SYBR Green RT-PCR protocol with β-actin as a housekeeping gene, the relative expression of controls was regarded as 1. The protein expression measurements were performed by semi-dry Western blotting using GAPDH as a loading control. The protein’s OD was expressed as a percentage of the target protein/GAPDH ratio. The bars represent the mean values (±standard deviation error bars, SD) of three independent experiments (*n* = 3) performed in triplicate and are presented relative to their control cells. The * marks the statistical significance of Glu treatments compared to control, # signs the statistical significance of combined (lutein with Glu) treatments compared to the Glu treatments at 48 h (*p* < 0.05).

**Figure 11 antioxidants-11-02269-f011:**
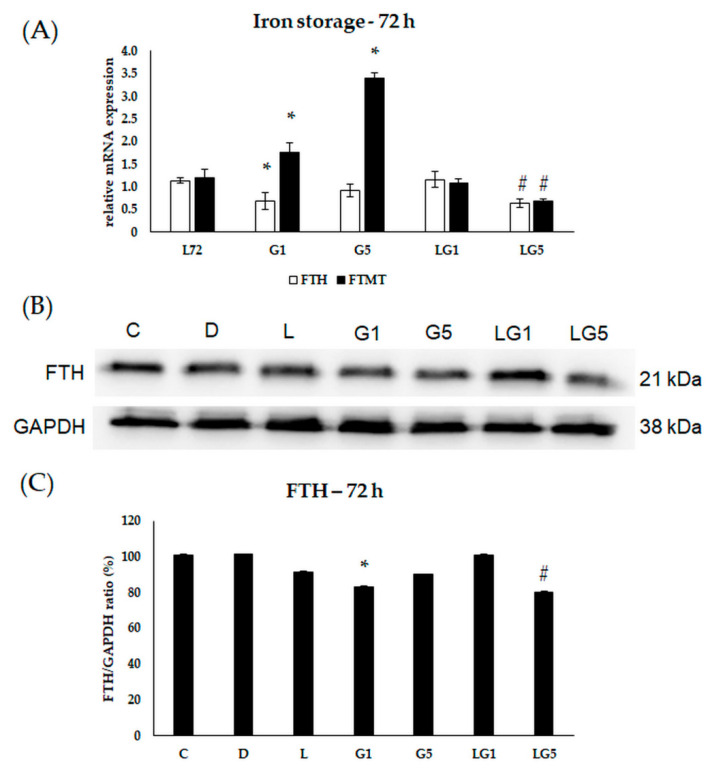
Determination of mRNA expression (**A**) and protein (**B**,**C**) levels of iron storing FTH and FTMT after lutein and Glu treatments of the SH-SY5Y cells at 72 h. SH-SY5Y cells were treated with 10 ng/µL of lutein, 1 mM, 5 mM Glu, or lutein with Glu together to detect the changes in the expression of FTH and FTMT. The gene expression measurements were performed by SYBR Green RT-PCR protocol with β-actin as a housekeeping gene, the relative expression of controls was regarded as 1. The protein expression measurements were performed by semi-dry Western blotting using GAPDH as a loading control. The protein’s OD was expressed as a percentage of the target protein/GAPDH ratio. The bars represent the mean values (±standard deviation error bars, SD) of three independent experiments (*n* = 3) performed in triplicate and presented relative to their control cells. The * marks the statistical significance of Glu treatments compared to control, # shows the statistical significance of combined (lutein with Glu) treatments compared to the Glu treatments at 72 h (*p* < 0.05).

**Figure 12 antioxidants-11-02269-f012:**
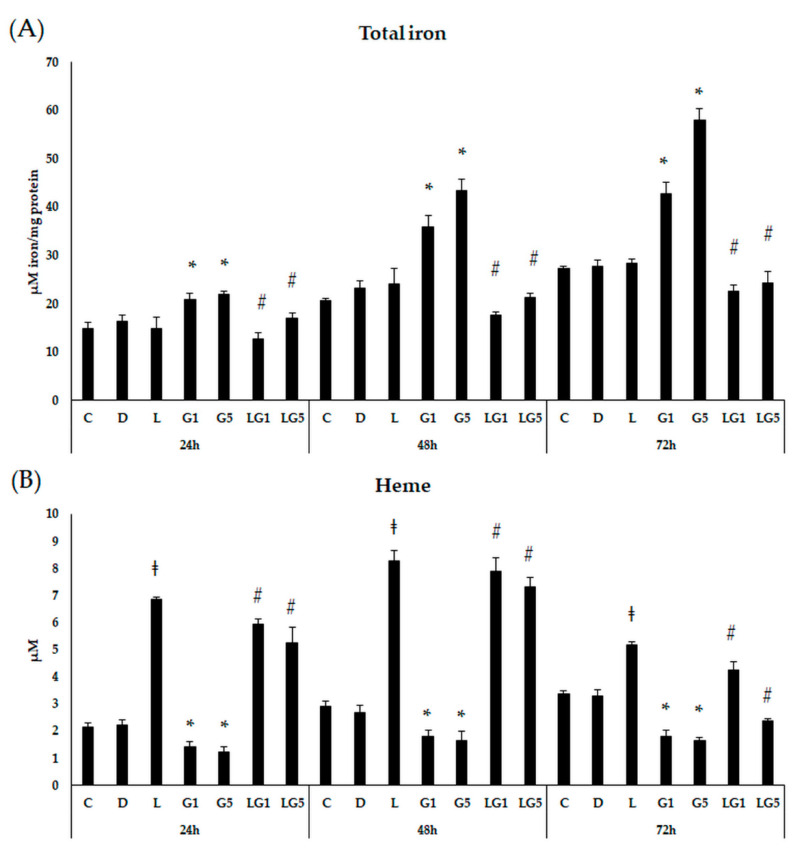
Determination of total iron levels (**A**) and heme concentrations (**B**) after lutein and Glu treatments of the SH-SY5Y cells. SH-SY5Y cells were treated with 10 ng/µL of lutein, 1 mM, 5 mM Glu, or lutein with Glu together to detect the changes in the iron content. The cells were incubated for 24 h, 48 h, and 72 h. A ferrozine-based colorimetric assay was used to perform the total iron measurements. The results of iron levels were normalised against the protein concentration and were expressed as µM iron/mg protein. For the heme measurements, a Heme Assay kit was used. The concentration of heme was expressed as µM. The bars represent the mean values (±standard deviation error bars, SD) of three independent experiments (*n* = 3) performed in triplicate and are presented relative to their control cells. The * marks the statistical significance of Glu treatments compared to control, ‡ signs the statistical significance of lutein treatments compared to control, # shows the statistical significance of combined (lutein with Glu) treatments compared to the Glu treatments at 24 h, 48 h and 72 h (*p* < 0.05).

**Figure 13 antioxidants-11-02269-f013:**
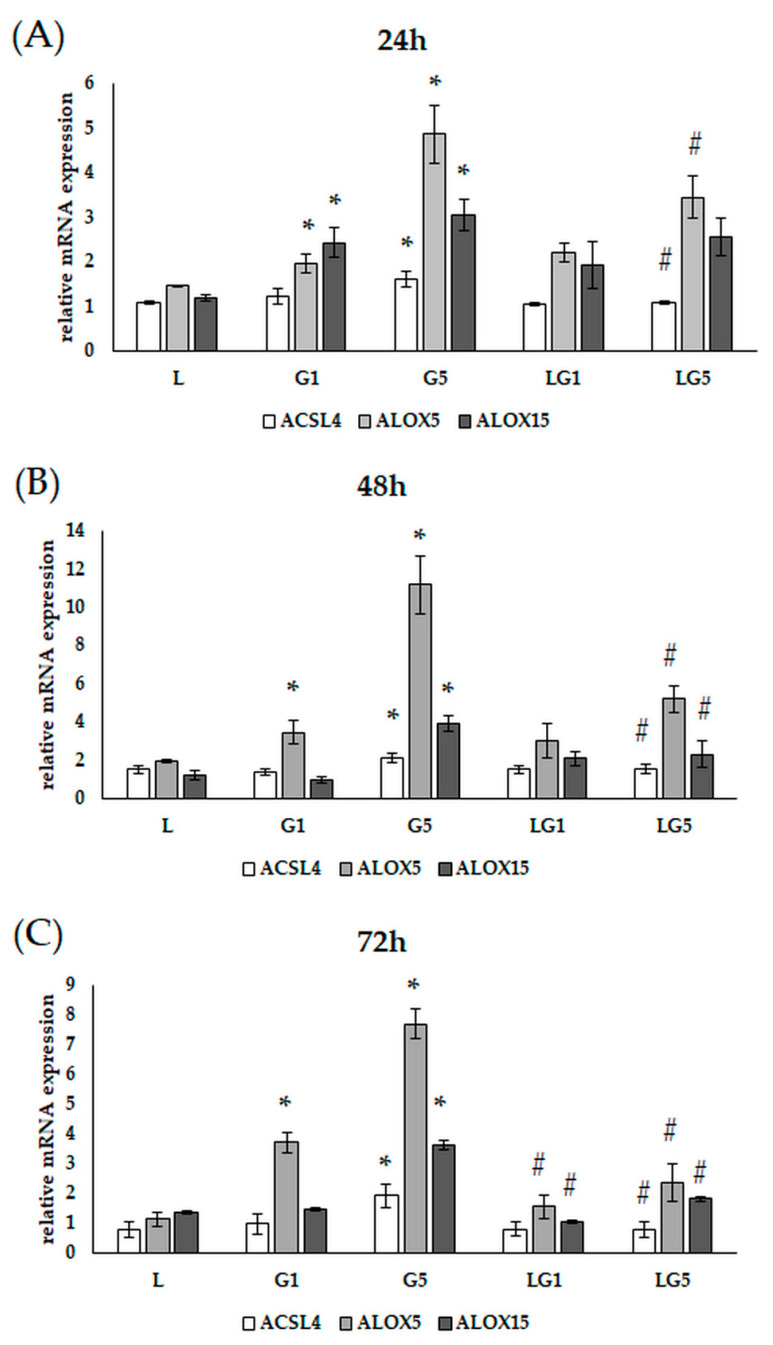
Determination of mRNA expression of ACSL4, ALOX5, and ALOX15 after lutein and Glu treatments of the SH-SY5Y cells at (**A**) 24 h (**B**) 48 h, and (**C**) 72 h. SH-SY5Y cells were treated with 10 ng/µL of lutein, 1 mM, 5 mM Glu, or lutein with Glu together to detect the changes in the expression of the examined genes. The gene expression measurements were performed by SYBR Green RT-PCR protocol using β-actin as a housekeeping gene. The relative expression of controls was regarded as 1. The bars represent the mean values (±standard deviation error bars, SD) of three independent experiments (*n* = 3) performed in triplicate and are presented relative to their control cells. The * marks the statistical significance of Glu treatments compared to the control, # shows the statistical significance of combined (lutein with Glu) treatments compared to the Glu treatments at 24 h, 48 h, and 72 h (*p* < 0.05).

**Table 1 antioxidants-11-02269-t001:** Abbreviations of the treatments.

Controls	Treatments
C-absolute control	Untreated
D2.5-DMSO equivalent to lutein	2.5 ng/µL of lutein
D5-DMSO equivalent to lutein	5 ng/µL of lutein
D7.5-DMSO equivalent to lutein	7.5 ng/µL of lutein
D10-DMSO equivalent to lutein	L-10 ng/µL of lutein
C1-DMSO equivalent to glutamate	G1-1 mM glutamate
C2-DMSO equivalent to glutamate	G2-2 mM glutamate
C3-DMSO equivalent to glutamate	G3-3 mM glutamate
C4-DMSO equivalent to glutamate	G4-4 mM glutamate
C5-DMSO equivalent to glutamate	G5-5 mM glutamate
C6-DMSO equivalent to glutamate	G6-6 mM glutamate
C7-DMSO equivalent to glutamate	G7-7 mM glutamate
C8-DMSO equivalent to glutamate	G8-8 mM glutamate
C9-DMSO equivalent to glutamate	G9-9 mM glutamate
C10-DMSO equivalent to glutamate	G10-10 mM glutamate
C15-DMSO equivalent to glutamate	G15-15 mM glutamate
C20-DMSO equivalent to glutamate	G20-20 mM glutamate

**Table 2 antioxidants-11-02269-t002:** Real-time PCR gene primer sequence list.

Target Gene	Gene Accession Number	Sequence 5′ → 3
β-actin forward	NM_007393.5	AGAAAATCTGGCACCACACC
β-actin reverse		GGGGTGTTGAAGGTGTCAAA
ACSL4 forward	NM_004458.3	TCTTGCTTTACCTATGGCTG
ACSL4 reverse		CAGTACAGTCTCCTTTGCTT
ALOX5 forward	NM_001256154.3	CGCGGTGGATTCATACG
ALOX5 reverse		GTCTTCAGCGTGATGTACT
ALOX15 forward	NM_001140.5	GAGGAGGAGTATTTTTCGGG
ALOX15 reverse		AATTTCCTTATCCAGGGCAG
DMT1 forward	NM_001174125.2	GTGGTTACTGGGCTGCATCT
DMT1 reverse		CCCACAGAGGAATTCTTCCT
FPN forward	NM_014585.6	AAAGGAGGCTGTTTCCATAG
FPN reverse		TTCCTTCTCTACCTTGGTCA
FTH forward	NM_002032.3	GAGGTGGCCGAATCTTCCTTC
FTH reverse		TCAGTGGCCAGTTTGTGCAG
FTMT forward	NM_177478.2	AAGGTGACCCCCATTTGTGC
FTMT reverse		GGGGCCCCCATCTTCACTAA
HAMP forward	NM_021175.4	CAGCTGGATGCCCATGTT
HAMP reverse		TGCAGCACATCCCACACT
TfR1 forward	NM_003234.4	CATGTGGAGATGAAACTTGC
TfR1 reverse		TCCCATAGCAGATACTTCCA

## Data Availability

Data is contained within the article and Appendix A.

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
