# Peer review of "Lutein Decreases Inflammation and Oxidative Stress and Prevents Iron Accumulation and Lipid Peroxidation at Glutamate-Induced Neurotoxicity"

_antioxidants, 2022, doi:10.3390/antiox11112269_

Round 1

Author Response

The present study investigated the neurotoxic consequences including oxidative
stress, inflammation, and iron metabolism of glutamate, as well as the
neuroprotective activities of lutein against glutamate using SH-SY5Y neuroblastoma
cells.
The full words can be added instead of abbreviation in abstract section such as SOD,
TNFα, IL-6, and IL-8.

Thank you for your comment! The full words were added to the lines 23-25, in the Abstract section.

First sentence in introduction. I would suggest adding a reference.‘’ Lutein is a
xanthophyll carotenoid, which has been widely used as supplementation because of
its antioxidant and anti-inflammatory benefits’’

Thank you for your comment! The following reference was added to the line 37, in Introduction.  

Buscemi S, Corleo D, Di Pace F, Petroni ML, Satriano A, Marchesini G. The Effect of Lutein on Eye and Extra-Eye Health. Nutrients. 2018 Sep 18;10(9):1321. doi: 10.3390/nu10091321. PMID: 30231532; PMCID: PMC6164534.

Here as well I would add a reference‘’ Glutamate, the crucial excitatory
neurotransmitter in the central nervous system (CNS), plays a key role in regulating
brain function’’

Thank you for your comment! The following references were added to the line 43, in Introduction.

Zhou Y, Danbolt NC. Glutamate as a neurotransmitter in the healthy brain. J Neural Transm (Vienna). 2014 Aug;121(8):799-817. doi: 10.1007/s00702-014-1180-8. Epub 2014 Mar 1. PMID: 24578174; PMCID: PMC4133642.

Reiner A, Levitz J. Glutamatergic Signaling in the Central Nervous System: Ionotropic and Metabotropic Receptors in Concert. Neuron. 2018 Jun 27;98(6):1080-1098. doi: 10.1016/j.neuron.2018.05.018. PMID: 29953871; PMCID: PMC6484838.

Instead for repeating the word full ‘’Glutamate’’ in all section I would suggest adding
abbreviated form ex..Glutamate (Glu) and abbreviated form all whole the text.

Thank you for your advice! The word „glutamate” was changed to the abbreviated form „Glu” in the manuscript. 

Introduction
Line 53--- full words.. when mentioned for the first time ‘’ TNFα, IL-6, IL-1β, INFγ
and IL-8’’

We apologise for the mistakes! The full names were added to the lines 52-53, in Introduction. 

Lines 57&58 ---I would add reference‘’ Iron is an essential element that plays a
fundamental role in the regulation and development of the CNS, besides participating
in several neuronal functions.’’

Thank you for your comment! The following reference was added to the line 59, in Introduction.

Singh N, Haldar S, Tripathi AK, Horback K, Wong J, Sharma D, Beserra A, Suda S, Anbalagan C, Dev S, Mukhopadhyay CK, Singh A. Brain iron homeostasis: from molecular mechanisms to clinical significance and therapeutic opportunities. Antioxid Redox Signal. 2014 Mar 10;20(8):1324-63. doi: 10.1089/ars.2012.4931. Epub 2013 Aug 15. PMID: 23815406; PMCID: PMC3935772.

Could you add the full names ‘’ ALOX5, ALOX15 and ASCL4’’

Thank you for your comment! The full names were added to lines 62-64, in Introduction.

Material and methods
‘’KOH’’ in I would add the full name.

Thank you for your comment! Potassium hydroxide was added to line 76, in Materials and methods 2.1.

In cell viability assay The abbreviation of treatments could be arranged in table.

Thank you for your comment! The abbreviations of treatments are represented in Table 1, in Materials and methods 2.3.

Could you please check all the abbreviations ‘’ full words.. when mentioned for the
first time’’

We apologise for the mistakes! The whole manuscript was thoroughly checked for the full words and abbreviations.

Line 113 Control- untreated absolute control keep this constant with above is it C or
the full word ‘’control’’?

Thank you for your comment! The expression „untreated absolute control” was changed in the manuscript to „control”.

Make sure all 2 are subscripted in ‘’CO2’’ and ‘’H2O2’’ as well as the numbers are all
superscripted as such in cell numbers‘’ 5x103’’ or , ‘’1x106 ‘’ cells or flask size ‘’ cm2’’
Any chemical compounds can be full names when mentioned for first time and make
sure for the formatting of number ‘’ subscripted or superscripted ’’ for example
KMnO4, FeCl3

Thank you for your comment! The full names were added to the chemical compounds in the manuscript (Materials and methods, lines 76, 147, 214-215, 243, 245-246, 251).

We apologise for the mistakes! The subscripted or superscripted numbers were formatting in the whole text.

In material and methods section --measurements of heme concentration
Cab be written as distilled water (dH2O) keep it constant with ‘’Measurements of
total iron’’ section when it mentioned for first time full words then abbreviated form
throughout the manuscript.

Thank you for your comment! The abbreviation dH2O was added into the manuscript after the first mention of distilled water (Materials and methods 2.10., line 246).

Reviewer 2 Report

The proposed manuscript entitled: “Lutein decreases inflammation and oxidative stress and prevents iron accumulation and lipid peroxidation at glutamate-induced neurotoxicity” is well-written and has a clear rationale. The authors focused on the biological activity of xanthophyll carotenoid lutein against glutamate-induced neurotoxicity. Lutein has been widely used as supplementation due to its protective effects in light-induced oxidative stress, so it seems to be a good idea to verify its potential neuroprotective and anti-inflammatory effects in a well-known in vitro injury model. 

The SH-SY5Y cell line is a tumor cell line, however frequently used as a model in much neurobiological research. In the literature, many authors claim that  these cells must be differentiated and show a phenotype characteristic of neurons. In this study, the authors unfortunately only assessed the effect of lutein on tumour cells. Would the authors comment on it?

Minor typographical errors were found throughout the manuscript and should be amended.

-          They are for subscript, superscript, the position of the space, and symbols ° (lines 87, 88, 137, 152, 154 and many more)- The Authors have to check the manuscript carefully.

-          Avoid starting sentences with a number or abbreviation (lines 135 and so on).

-          The Figure descriptions are too long. The Authors repeat all the information described earlier in the methods. I would suggest making these descriptions shorter. Since the statistical analysis was described in section 2.12, it is not necessary to repeat all the information below the figures.

-          Line 350- It seems that The Authors can use SOD because the Abbreviation was defined.

Author Response

The proposed manuscript entitled: “Lutein decreases inflammation and oxidative stress and prevents iron accumulation and lipid peroxidation at glutamate-induced neurotoxicity” is well-written and has a clear rationale. The authors focused on the biological activity of xanthophyll carotenoid lutein against glutamate-induced neurotoxicity. Lutein has been widely used as supplementation due to its protective effects in light-induced oxidative stress, so it seems to be a good idea to verify its potential neuroprotective and anti-inflammatory effects in a well-known in vitro injury model. 

The SH-SY5Y cell line is a tumor cell line, however frequently used as a model in much neurobiological research. In the literature, many authors claim that  these cells must be differentiated and show a phenotype characteristic of neurons. In this study, the authors unfortunately only assessed the effect of lutein on tumour cells. Would the authors comment on it?

Thank you for your question!

Both undifferentiated and differentiated SH-SY5Y cells have been utilized for in vitro experiments requiring neuronal like cells.

The effect of antioxidant or neuroprotective extracts, ingredients or derivatives have been examined on differentiated SH-SY5Y cells (https://doi.org/10.3390/ijms13089692; https://doi.org/10.1016/j.neulet.2022.136956; https://doi.org/10.3390/biom10111530) and on undifferentiated SH-SY5Y cells, more frequently the latter (https://doi.org/10.1016/j.jep.2022.115836; https://doi.org/10.1016/j.fitote.2022.105346; https://doi.org/10.3892/ijmm.2019.4139; https://doi.org/10.1016/j.bioorg.2022.106179; https://doi.org/10.1155/2012/728342; https://doi.org/10.1007/s13197-022-05575-1; https://doi.org/10.1007/s11130-022-01004-y; https://doi.org/10.1007/s12640-022-00601-8; https://doi.org/10.1016/j.ejphar.2022.175307; https://doi.org/10.1007/s11033-022-08039-z).

Literatures have been reported the effect of glutamate, including glutamate-induced neurotoxicity or oxidative stress on both differentiated (https://doi.org/10.3390/biom10111530; https://doi.org/10.1155/2014/674164) and undifferentiated SH-SY5Y cells (https://doi.org/10.1007/s12264-010-0813-7; https://doi.org/10.1016/j.ijpharm.2022.121774; https://doi.org/10.1016/j.heliyon.2021.e07310; https://doi.org/10.1016/j.jad.2021.07.054; https://doi.org/10.1002/ptr.7057;  https://doi.org/10.1016/j.neuro.2021.02.006).

Upon differentiation with retinoic acid and 12-O-tetradecanoylphorbol 13-acetate SH-SY5Y cells has been reported to increase the production of neurotrophic factors and upregulate the anti-apoptotic Bcl-2 family member protein (https://doi.org/10.1046/j.1471-4159.1996.67010131.x; https://doi.org/10.1046/j.1471-4159.2000.0750991.x; https://doi.org/10.1146/annurev.pharmtox.38.1.289). Neurotrophins are known to have neuroprotective, neurorestorative and stimulatory effects on diseased neurons also have reported to downregulate oxidative stress via reducing ROS (https://doi.org/10.1007/s00018-022-04397-w; https://doi.org/10.1016/j.expneurol.2021.113901; https://doi.org/10.1016/j.expneurol.2012.09.001; https://doi.org/10.1016/j.bbr.2012.10.047;  https://doi.org/10.1002/dneu.20765; https://dx.doi.org/10.4331/wjbc.v1.i5.133)

In our study, we focused on the effect of lutein on ROS, inflammation, and neuroprotection upon glutamate-induced neurotoxicity. According to the findings in the literature, thus differentiated SH-SY5Y cells alter the anti-apoptotic regulation and inflammatory and oxidative stress response, we decided to perform our experiments on undifferentiated SH-SY5Y cells. 

Minor typographical errors were found throughout the manuscript and should be amended.

-          They are for subscript, superscript, the position of the space, and symbols ° (lines 87, 88, 137, 152, 154 and many more)- The Authors have to check the manuscript carefully.

Our apologise for the mistakes! The subscripted or superscripted numbers were formatted in the whole manuscript, the position of space and symbols ° were corrected and unified.

-          Avoid starting sentences with a number or abbreviation (lines 135 and so on).

Thank you for your comment! Sentences in lines (without display the tracked changes) 125, 126, 127, 153 were paraphrased.

-          The Figure descriptions are too long. The Authors repeat all the information described earlier in the methods. I would suggest making these descriptions shorter. Since the statistical analysis was described in section 2.12, it is not necessary to repeat all the information below the figures.

Thank you for your comment! The statistical description was shortened below the figures.

-          Line 350- It seems that The Authors can use SOD because the Abbreviation was defined.

Thank you for your comment! The name “Superoxide dismutase” was replaced with the abbreviated form SOD.  

Reviewer 3 Report

The abbreviations have to be reported in extenso at their first appearance in the text.

Please check the use of superscripts and subscripts

The structure of lutein could be included as figure.

The quality of the figures seems to be low. Please replace with figures with high resolution.

The activity of CAT and SOD could be better explained using the CAT/SOD ratio that is an index of anitoxidant activity.

In figure captions it is not clear the level of statistical significance. Authors used different symbols without explaining the meaning of the statistical significance of each one.

Author Response

The abbreviations have to be reported in extenso at their first appearance in the text.

Our apologise for the mistakes! The manuscript was thoroughly checked, and the abbreviations were written in full names at the first mention.

Please check the use of superscripts and subscripts

Our apologise for the mistakes! The subscripted or superscripted numbers were checked and formatted in the manuscript.

The structure of lutein could be included as figure.

Thank you for your comment! The structure of lutein was added to the manuscript and is represented in Figure 1.

The quality of the figures seems to be low. Please replace with figures with high resolution.

Thank you for your comment! The size of the figures is increased in the manuscript. The recommendation of MDPI’s layout style guide is a minimum resolution of 600 dpi for the figures. Our figures are in a resolution of 600 dpi with 1700 pixels height or width and in format tif. Maybe due to the style “narrowed width due to the side indentation” of the paper the figures seem to be low resolution.

The activity of CAT and SOD could be better explained using the CAT/SOD ratio that is an index of anitoxidant activity.

Thank you for your comment! Figure 3 was completed with the SOD/CAT ratio and represented in Figure 3C.

In figure captions it is not clear the level of statistical significance. Authors used different symbols without explaining the meaning of the statistical significance of each one.

The materials and methods section 2.12. was added with the text “in each analysis”: The statistical significance was set at p-value <0.05 in each analysis. The * indicates p<0.05 between glutamate treatment and control cells, # indicates p<0.05 between glutamate treatment and lutein with glutamate treatment, Ç‚ indicates p<0.05 between lutein treatment and control cells.
